# Blood-brain barrier breakdown in non-enhancing multiple sclerosis lesions detected by 7-Tesla MP2RAGE $\Delta T_1$ mapping

**Seongjin Choi** [1], **Margaret Spini**[2], **Jun Hua**[3,4], **Daniel M. Harrison** [1,5]*

1 Department of Neurology, University of Maryland School of Medicine, Baltimore, Maryland, United States of America, 2 University of Maryland School of Medicine, Baltimore, Maryland, United States of America, 3 Department of Radiology and Radiological Sciences, Johns Hopkins University School of Medicine, Baltimore, Maryland, United Stated of America, 4 F.M. Kirby Research Center for Functional Brain Imaging, Kennedy Krieger Institute, Baltimore, Maryland, United States of America, 5 Department of Neurology, Johns Hopkins University School of Medicine, Baltimore, Maryland, United States of America

* dharrison@som.umaryland.edu

## Abstract

Although the blood-brain barrier (BBB) is altered in most multiple sclerosis (MS) lesions, gadolinium enhancement is seen only in acute lesions. In this study, we aimed to investigate gadolinium-induced changes in $T_1$ relaxation time in MS lesions on 7-tesla (7T) MRI as a means to quantify BBB breakdown in non-enhancing MS lesions. Forty-seven participants with MS underwent 7T MRI of the brain with a magnitude-prepared rapid acquisition of 2 gradient echoes (MP2RAGE) sequence before and after contrast. Subtraction of pre- and post-contrast $T_1$ maps was used to measure $T_1$ relaxation time change ($\Delta T_1$) from gadolinium. $\Delta T_1$ values were interrogated in enhancing white matter lesions (ELs), non-enhancing white matter lesions (NELs), and normal appearing white matter (NAWM) and metrics were compared to clinical data. $\Delta T_1$ was measurable in NELs (median: -0.139 (-0.304, 0.174) seconds; p < 0.001) and was negligible in NAWM (median: -0.001 (-0.036, 0.155) seconds; p = 0.516). Median $\Delta T_1$ in NELs correlated with disability as measured by Expanded Disability Status Scale (EDSS) (rho = -0.331, p = 0.026). Multiple measures of NEL $\Delta T_1$ variability also correlated with EDSS. NEL $\Delta T_1$ values were greater and more variable in patients with progressive forms of MS and greater in those not on MS treatment. Measurement of the changes in $T_1$ relaxation time caused by contrast on 7T MP2RAGE reveals clinically relevant evidence of BBB breakdown in NELs in MS. This data suggests that NEL $\Delta T_1$ should be evaluated further as a potential biomarker of persistently disrupted BBB in MS.

## Introduction

Gadolinium enhancement of white matter lesions (WMLs) has long been used as a surrogate marker of active inflammation and blood-brain barrier (BBB) breakdown in multiple sclerosis (MS) [1]. The presence of enhancing WMLs can be used clinically to meet diagnostic criteria

**Data Availability Statement:** Data cannot be shared publicly because of potentially identifying information. Data may be made available to researchers who meet criteria for access to

confidential data by arranging a data transfer agreement with the University of Maryland, Baltimore. Requests for such an agreement can be initiated by contacting the Sponsored Projects office at the University of Maryland Baltimore at team-red@ordmail.umaryland.edu.

**Funding:** This work was funded by grants from the NIH - National Institute of Neurological Disorders and Stroke (NINDS 1K23NS072366-01A1 and 1R01NS104403-01; PI: DMH) and EMD Serono, Inc (PI: DMH). The funders had no role in study design, data collection and analysis, decision to publish, or preparation of the manuscript. https://www.ninds.nih.gov/ https://www.emdserono.com/us-en/company.html.

**Competing interests:** I have read the journal's policy and the authors of this manuscript have the following competing interests: DMH has received consulting fees from EMD Serono, Inc., Genentech, Sanofi-Genzyme, and Biogen.

for dissemination in time [2] and as a marker of relapsing disease activity [3]. Reductions in the number of enhancing WMLs on MRI has been used as a treatment effect outcome measure in most clinical trials [4], and the availability of this tool has arguably led to more rapid development of MS therapeutics.

While gadolinium-enhanced MRI is a useful tool for differentiation of acutely inflamed WMLs from those that are not, the inflammatory status of non-enhancing lesions (NELs) are more difficult to elucidate. Although no longer practical due to toxicity concerns, early triple-dose gadolinium studies demonstrated an increase in the number of enhancing lesions visualized compared to standard-dose gadolinium in MS patients [5, 6]. Further, enhancing lesions were found in 42% of patients with primary progressive MS (PPMS)–a subtype of MS typically not associated with acute relapsing activity [7]. It is still unclear if these findings represent milder acutely inflamed lesions or more chronic disruptions of the BBB in older, possibly chronically inflamed WMLs. Regardless, it is clear that qualitative evaluation of WML contrast enhancement is an insufficient method to fully characterize WMLs and more quantifiable methods are needed.

Measurements of vascular permeability by dynamic contrast-enhanced (DCE) MRI are a candidate method for measuring BBB breakdown in MS. In fact, prior DCE MRI studies suggest increased permeability in non-enhancing WMLs and NAWM in patients with MS [8]. Alterations in $T_1$-relaxation time, which is shortened in the presence of gadolinium [9], have also been used as a measure of BBB integrity. Multiple studies have found $T_1$ shortening in both enhancing and non-enhancing WMLs [10].

Despite the clear potential for these techniques to measure BBB integrity, their clinical application has been limited by practical considerations. For example, permeability measurements require complex post-acquisition processing, which make this impractical for use in large studies or in the clinic. Echo-planar imaging (EPI) techniques for measurement of $T_1$ relaxation time suffer from distortions and partial volume averaging that reduce the accuracy of $T_1$ measurements in small structures. Spoiled gradient-echo techniques for $T_1$ measurement are more robust but require multiple acquisitions that must be co-registered before pre and post contrast images can be compared, which increases scan time and decreases the accuracy and reproducibility of measurement.

More recently, the magnetization-prepared rapid acquisition of 2 gradient echoes (MP2RAGE) technique has been used for $T_1$ relaxation time measurement [11]. This technique, which produces both a $T_1$-weighted ($T_1$-w) MPRAGE-like image and a quantitative $T_1$ map, provides several advantages over previous methods. MP2RAGE acquires all necessary images in a simultaneous acquisition, and thus co-registration is not required. High spatial resolutions are possible in clinically acceptable scan times without partial volume averaging effects. MP2RAGE images are also inherently B1 field inhomogeneity corrected, making them ideal for application at higher magnetic fields [11]. The promise of MP2RAGE as a useful tool in the study of MS has already been evaluated, with early results showing robust and reproducible $T_1$ measurements that correlate with disability and increases in the ability to visualize white and gray matter lesions [12, 13].

In this study, we aimed to take advantage of MP2RAGE's ability to provide robust, high resolution maps of $T_1$ relaxation time to measure gadolinium-induced $T_1$ signal change as a biomarker of BBB leakage in non-enhancing lesions in MS. We looked to evaluate this at 7T, where images can be acquired at fine resolution and the absolute $T_1$ relaxation time difference between tissues are greater [13, 14], further enhancing the potential accuracy of measurement over prior, similar studies.

## Materials and methods

### Participants

Volunteers aged 18 to 65 with diagnoses of relapsing-remitting MS (RRMS), secondary progressive MS (SPMS), and progressive MS (PPMS) according to revised 2010 McDonald Criteria [2] were recruited from the Johns Hopkins Multiple Sclerosis Center and the University of Maryland Center for Multiple Sclerosis Treatment and Research. Participants were excluded for contraindications to MRI (i.e. metallic foreign bodies) or gadolinium contrast (i.e. previous allergy to contrast, renal failure). Demographic and clinical characteristics of the study population is provided in Table 1.

### MRI acquisition

All participants underwent MRI of the whole brain on a 7T MRI scanner (Philips, Achieva, The Netherlands) with a volume transmit head coil and a 32-channel receiver coil (Nova Medical Inc.) as part of a multi-modal brain MRI study. MP2RAGE and magnetization prepared fluid attenuated inversion recovery (MPFLAIR) sequences were used in this analysis. MP2RAGE sequence parameters are as follows: MP2RAGE TR = 8500 ms, TR = 6.9 ms, TE = 2.1 ms, inversion times = 1000/3000 ms, flip angles = 5/5 degrees, Turbo factor = 252, Field-of-view = 220x220 mm$^2$, near-isotropic resolution of 0.7x0.688x0.688 mm$^3$, SENSE

**Table 1. Demographic and clinical characteristics of the study population.**

| Characteristic | Full Cohort (n = 47) | Relapsing-Remitting MS Subjects (n = 34, 72.3%) | Progressive MS Subjects (n = 13, 27.7%) |
|---|---|---|---|
| Age, mean (SD) in years | 46.2 (11.2) | 44.3 (11.4) | 51.2 (9.4) |
| Sex, female (%) | 30 (63.8%) | 23 (67.7%) | 7 (53.9%) |
| Symptom duration, mean (SD) in years | 11.5 (8.3) | 11.4 (8.8) | 11.9 (7.2) |
| On MS treatment (%) | 34 (72.3%) | 27 (79.4%) | 7 (53.9%) |
| Interferon-beta | 6 (12.8%) | 4 (11.8%) | 2 (15.4%) |
| Glatiramer acetate | 7 (14.9%) | 7 (20.6%) | 0 (0%) |
| Natalizumab | 4 (8.5%) | 3 (8.8%) | 1 (7.7%) |
| Teriflunomide | 1 (2.1%) | 1 (2.9%) | 0 (0%) |
| Fingolimod | 4 (8.5%) | 3 (8.8%) | 1 (7.7%) |
| Dimethyl Fumarate | 11 (23.4%) | 9 (26.5%) | 2 (15.4%) |
| Rituximab | 1 (2.1%) | 0 (0%) | 1 (7.7%) |
| EDSS score, median (range) | 3.0 (1, 6.5) | 3.0 (1.0, 6.0) | 6.0 (4.0, 6.5) |
| SDMT # correct, mean (SD) | 53.2 (12.3) | 55.7 (12.8) | 46.5 (7.9) |
| PASAT # correct, mean (SD) | 44.8 (12.8) | 45.9 (13.4) | 42.1 (11.3) |
| MFIS score, mean (SD) | 39.5 (20.4) | 35.5 (21.6) | 49.9 (12.3) |
| 9HPTDOM, mean (SD) | 45.3 (111) | 25.3 (15.7) | 40.9 (21.3) |
| 9HPTNONDOM, mean (SD) | 32.3 (22.5) | 27.1 (18.6) | 46.1 (26.6) |
| 25FTW, mean (SD) | 8.66 (13.2) | 5.5 (3.4) | 17.0 (23.2) |
| # of WML, mean (SD) | 87.4 (68.5) | 77.6 (51.8) | 113.1 (98.1) |
| Volume of WML, mean (SD) in mm3 | 6099.8 (5790.4) | 5753.9 (5193.7) | 7004.3 (7288.5) |
| # of cases with enhancing lesions (%) | 5 (10.6%) | 4 (8.5%) | 1 (2.1%) |
| # of enhancing lesions seen in participants with enhancement, median (range) | 1 (1, 3) | 1 (1, 3) | 1 (1, 1) |

SD = standard deviation; # = number; MS = multiple sclerosis; RRMS = relapsing-remitting multiple sclerosis; PMS = progressive multiple sclerosis; EDSS = Expanded Disability Status Scale; SDMT = Symbol Digit Modalities Test; PASAT = Paced Auditory Serial Addition Test; MFIS = Modified Fatigue Impact Scale; 9HPTDOM = 9-hole peg test for dominant hand; 9HPTNONDOM = 9-hole peg test for non-dominant hand; 25FTW = timed 25-foot walk.

acceleration factor = 2x2, total acquisition time = 9 min 46 sec. Sequences were obtained before and approximately 10 minutes after intravenous gadolinium contrast agent (0.1 mmol/kg) administration. Detailed MPFLAIR sequence parameters are as follows: TR = 8000 ms, TE = 400 ms, TI = 2,077 ms, flip angle = 90 degrees, SENSE acceleration factor = 2x3, total acquisition time = 10 min 48 sec.

## Image processing

All MRI data were processed for $T_1$-w images and a quantitative $T_1$ relaxation time map ($T_1$ map), as previously described [11]. This was accomplished using custom software written in Matlab (Mathworks, Inc., Natick, MA), which was based on publicly available code provided by the developers of this sequence (https://github.com/JosePMarques/MP2RAGE-related-scripts). A denoised $T_1$-w image was created by multiplying the second inversion image (after N4 bias correction) [15] and the $T_1$-w image (after adding 0.5 to make its intensity range positive) to suppress the background noise of MP2RAGE $T_1$-w image [16]. The corresponding denoised $T_1$-w image was used for skull stripping and co-registration. After $T_1$ map processing and linear registration of post-contrast images to the pre-contrast space, $T_1$ difference maps, which we term "delta $T_1$" ($\Delta T_1$) maps, were generated by subtracting pre-contrast from post-contrast $T_1$ maps. Masks were manually created for all WMLs by reviewing MPFLAIR and MP2RAGE images in tandem; MPFLAIR images were used as guidance and masks drawn on the pre-contrast $T_1$ map. Our process for WML masking on 7T MP2RAGE is previously described in Spini et al (2020) [17]. Masks were separately drawn for lesions with visually apparent contrast enhancement on $T_1$-w images. Non-enhanced lesion (NEL) masks were created by subtraction of enhancing lesions (ELs) from the overall WML mask (example in Fig 1).

A custom processing pipeline was created in the Java Image Science Toolkit (JIST, version 3.0, https://www.nitrc.org/projects/jist) [18] environment using tools from the CBS Tools processing package for high resolution, ultra-high field MRI [19] and the Lesion-TOADS segmentation algorithm [20] for brain segmentation. Segmentation was performed after lesion filling based on manually drawn WML masks. A normal-appearing white matter (NAWM) mask was created by subtracting WML masks from cerebral WM masks. SC, MS, DMH delineated

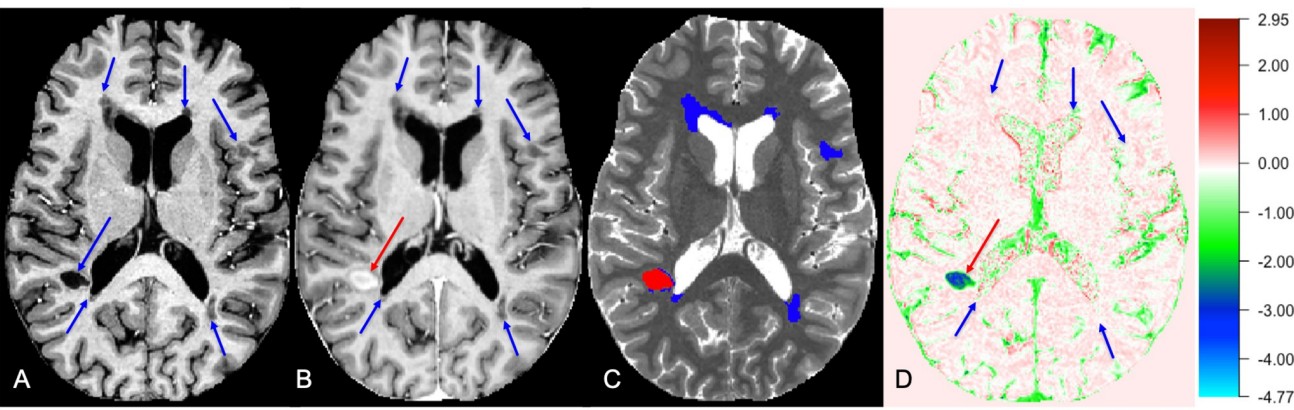

**Fig 1. Lesion masking for ELs and NELs as seen on pre- and post-contrast $T_1$w images, $T_1$ map, and $\Delta T_1$ map.** A: Lesioned areas (indicated by blue arrows) appeared hypointense on pre-contrast $T_1$-weighted image. B: EL area (indicated by a red arrow) on post-contrast $T_1$-weighted image while NELs remains unenhanced. C: White matter lesions seen on $T_1$ map (EL in red, NEL in blue). D: White matter lesions seen on $\Delta T_1$ map (EL in red, NEL in blue, units: s). The color bar displays the full range of $\Delta T_1$ (units: s) values only in this participant. EL = enhancing lesion; NEL = non-enhancing lesion; $T_1$w = $T_1$-weighted.

WML and DMH reviewed and finalized them. SB delineated EL, and DMH reviewed and finalized them. SC counted lesions, and DMH reviewed and finalized them.

### Disability measures

Disability was characterized through use of the Kurtzke Expanded Disability Status Scale (EDSS) score [21]. Upper extremity function was measured using the 9-hole peg test (9HPT) and gait function was assessed through the timed 25-foot walk (25FTW) [22]. The Symbol Digit Modalities Test (SDMT) and paced auditory serial addition test (PASAT) were used to assess cognitive functioning [22]. The modified fatigue impact scale (MFIS) was used to assess MS-related fatigue [23].

### Statistical analysis

Differences between pre-contrast and post-contrast $T_1$ metrics were assessed by paired t-test and matched pairs Wilcoxon test. Group differences for EDSS, MS phenotype, and treatment effect were assessed by Mann-Whitney U-test. To allow for statistical analysis using dichotomous variables, we divided subjects into groups of $\leq$ or $>$ the median EDSS score, which would compare mild and moderate disability. We also performed a logistic regression model with adjustment for MS phenotype to ensure any treatment effect findings were not influenced by MS phenotype (due to propensity for progressive patients to be untreated). Correlations were assessed by Spearman or Pearson correlations, as appropriate. R (version 4.0.3, R Core Team, Vienna, Austria, https://www.R-project.org/) was used to perform all statistical analysis. Python (version 3.7.1) was used to visualize data in part. P-values lower than 0.05 were considered statistically significant in all analyses. The false discovery rate (FDR) method was used to correct for multiple comparisons. However, as overcorrection for false discovery can often mask true positive findings in small cohorts (such as this study) and inhibit hypothesis-generating research, we have reported both unadjusted and FDR-adjusted p-values to allow readers to come to their own conclusions [24].

### Results

Demographics of the recruited cohort are shown in Table 1. There were forty-seven (47) participants with MS, of which 34 (72.3%) were of the RRMS phenotype and 13 (27.7%) were of progressive phenotypes (SPMS or PPMS). Most subjects in this cohort (72.3%) were on disease modifying therapies at the time of their scan. Visual review of scans in this cohort revealed only 5 participants (10.6%) with visually enhancing WMLs, with a range of 1–3 enhancing lesions on those scans.

   Table 2 shows data derived from $T_1$ maps. Consistent with prior literature [25], $T_1$ relaxation time (units: seconds) in MS lesions (median (range): 2.430 (2.097, 2.806)) was prolonged (P<0.001) on pre-contrast $T_1$ maps compared to NAWM (median (range): 1.300 (1.16, 1.403)). As expected, ELs showed a significantly shorter (P = 0.017) post-contrast $T_1$ relaxation times when we compared the means of per subject median. Although the absolute difference between the inter-subject median of pre- and post- contrast median $T_1$ in NAWM was very small, resulting in a $\Delta T_1$ of -0.004 (-0.035, 0.146), this difference was significant (p = 0.006). However, no appreciable difference was seen in comparing the inter-subject mean of NAWM $T_1$ (p = 0.411). Although NELs appeared unenhanced on $T_1$-w images, $T_1$ relaxation time was significantly shorter (P<0.001) in NELs on post-contrast $T_1$ maps (median (range): 1.784 (1.286, 2.591)) compared to pre-contrast (median (range): 1.912 (1.478, 2.681)). Consequently, an appreciable $\Delta T_1$ value was found for NELs (median (range): -0.134 (-0.281, 0.134)), which

**Table 2. $T_1$ metrics from pre- and post-contrast $T_1$ maps.**

| Tissue | Measure[*] | Median pre-contrast $T_1$ (units: s) | Median post-contrast $T_1$ (units: s) | Difference significance | Median $\Delta T_1$ (units: s) |
|---|---|---|---|---|---|
| WML | Mean (SD) | 1.938 (0.27) | 1.819 (0.274) | P < 0.001[a] | -0.129 (0.072) |
| | Median (range) | 1.915 (1.478, 2.681) | 1.784 (1.286, 2.591) | P < 0.001[b] | -0.135 (-0.281, 0.134) |
| NEL | Mean (SD) | 1.937 (0.27) | 1.820 (0.274) | P < 0.001[a] | -0.129 (0.072) |
| | Median (range) | 1.912 (1.478, 2.681) | 1.784 (1.286, 2.591) | P < 0.001[b] | -0.134 (-0.281, 0.134) |
| EL[**] | Mean (SD) | 2.444 (0.253) | 1.477 (0.544) | P = 0.017[a] | -0.957 (0.589) |
| | Median (range) | 2.430 (2.097, 2.806) | 1.217 (0.988, 2.069) | P = 0.063[b] | -1.100 (-1.604, -0.314) |
| NAWM | Mean (SD) | 1.290 (0.056) | 1.286 (0.057) | P = 0.411[a] | 0.004 (0.029) |
| | Median (range) | 1.300 (1.16, 1.403) | 1.290 (1.188, 1.461) | P = 0.006[b] | -0.004 (-0.035, 0.146) |
| cGM | Mean (SD) | 1.897 (0.055) | 1.805 (0.049) | P < 0.001[a] | -0.09 (0.041) |
| | Median (range) | 1.898 (1.666, 2.025) | 1.818 (1.68, 1.872) | P < 0.001[b] | -0.096 (-0.143, 0.097) |

[a]Paired t-test (two-tailed);

[b]Matched-pairs Wilcoxon test (two-tailed); SD = standard deviation; WML = white matter lesion; NEL = non-enhancing lesion; EL = enhancing lesion;

NAWM = normal-appearing white matter; cGM = cortical gray matter.

[*] All measures in the second column indicate mean and median values of 47 individual intra-subject median $T_1$ values collected within each tissue.

[**] Gd-enhanced lesions were only confirmed in five out of forty-seven participants, so this analysis is in only those 5 sujects.

was of a far greater magnitude than seen for NAWM. A small, but significant difference was also noted between pre- and post-contrast $T_1$ values in cortical gray matter (cGM).

To determine if central tendency measures for $\Delta T_1$ in NELs were influenced heavily by a small number of lesions on each scan or if this reflected a more diffuse process, heat maps of $\Delta T_1$ in NEL masks were visually reviewed for each subject (examples in Fig 2). In all cases, $\Delta T_1$ appeared to represent a scattered phenomenon, as none showed large deviations in $\Delta T_1$ focal to a small number of lesions along with a neutral $\Delta T_1$ in all other lesions.

$\Delta T_1$ values in NELs, NAWM, and cGM were also evaluated for their relationship with other measures of lesion severity. Per-subject mean and median $\Delta T_1$ did not correlate with mean and median pre-contrast $T_1$ (rho = -0.207, P = 0.16 and rho = -0.286, P = 0.05, respectively) in NELs. In full cohort, voxel-wise analysis, all $\Delta T_1$ and pre-contrast $T_1$ (Fig 3B) voxels in NELs were only weakly correlated (rho = -0.261, P<0.001; Fig 3B). Comparatively, $\Delta T_1$ and pre-contrast $T_1$ in all voxels in NAWM and cGM showed stronger correlations (rho = -0.348, P<0.001 in Fig 3E; rho = -0.368, P<0.001 in Fig 3H, respectively). Fig 3A, 3D and 3G show the diversity of distribution of $\Delta T_1$ in NEL, NAWM, and cGM, respectively, in the study cohort using overlaid kernel density estimation (KDE) and Fig 3C, 3F and 3I show the same pre-contrast $T_1$ for NEL, NAWM, and cGM. In general, pre-contrast $T_1$ showed a more diverse set of distributions, including many of which were bimodal and skewed, whereas $\Delta T_1$ was more normally distributed in all tissues with different extents of diversity. Compared to NAWM and GM, voxel-wise values in NELs had a wider range of $\Delta T_1$ and pre-contrast $T_1$ values. Magnitude and variability measures of NEL $\Delta T_1$ showed no significant correlation with overall or average WML lesion volume per subject (Table 3). The magnitude of NEL $\Delta T_1$ did not correlate with WML count, but NEL $\Delta T_1$ variance and IQR did correlate with WML count (rho = 0.582, P<0.001 and rho = 0.489, P<0.001, respectively). The correlation between lesion count and lesion $\Delta T_1$ variance and IQR were significant even after controlling for age, sex, and symptom duration and corrected for non-conservative multiple comparisons (false discovery rate) (S1 Table).

Correlation analysis was used to examine the clinical significance of gadolinium-induced $T_1$ shortening (Table 4). In bivariate correlation analysis, median $\Delta T_1$ in NELs correlated with

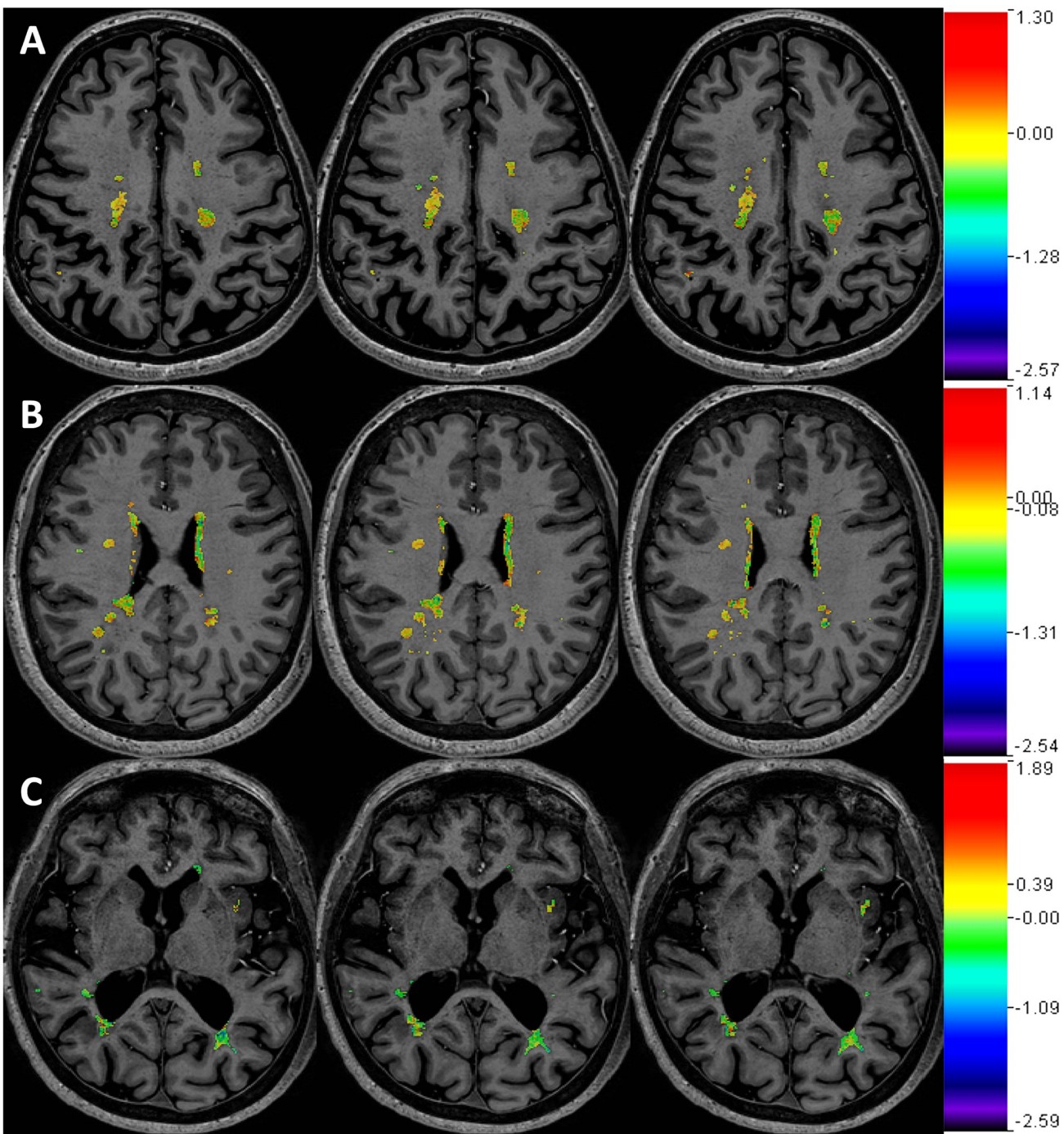

**Fig 2. Magnitude of $\Delta T_1$ in non-enhanced lesion areas.** Shown are images from three separate participants (A: SPMS/female/53, B: RRMS/female/52, C: SPMS/female/61). Color maps of $\Delta T_1$ (units: s) in NEL's were overlaid upon $T_1$-weighted images. Each color bar (not scaled) displays the full range of $\Delta T_1$ only within NEL of the corresponding participant. Three representative participants shown for display purposes (analysis was in all subjects). Each row shows three consecutive image slices of a single participant. SPMS = secondary progressive multiple sclerosis; RRMS = relapsing-remitting multiple sclerosis; BBB = blood-brain barrier; NEL = non-enhancing lesion.

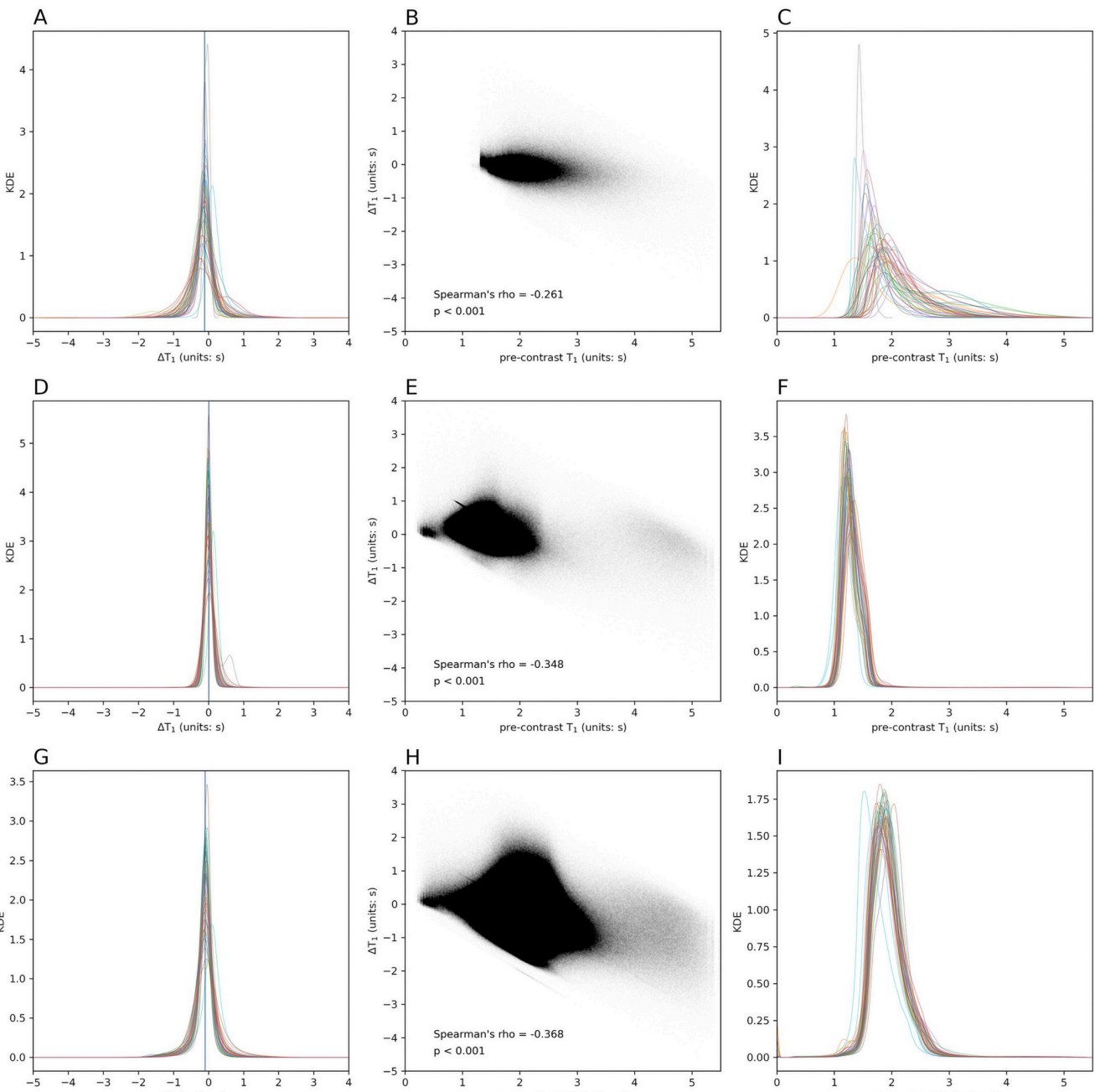

**Fig 3. $\Delta T_1$ and Pre-contrast $T_1$ in NEL, NAWM, cGM of entire cohort. A**: Overlaid kernel density estimations (KDEs) of $\Delta T_1$ in NEL. Each colored line represents the smoothed histogram using KDE for all voxels in one subject. Blue vertial line indicates voxel-wise mean $\Delta T_1$ value (-0.116 s), and green line denotes voxel-wise median $\Delta T_1$ value (-0.105 s). **B**: Per voxel Spearman's rho coefficient for the relationship of pre-contrast $T_1$ and $\Delta T_1$ of all NEL voxels of the entire cohort shows a weak correlation, which indicates that observed $\Delta T_1$ is not solely driven by initial $T_1$ before contrast agent administration. **C**: Overlaid KDEs of pre-contrast $T_1$ in NEL **D**: Overlaid KDEs of $\Delta T_1$ in NAWM. **E**: Per voxel Spearman's rho coefficient in NAWM shows a stronger correlation than NEL (blue line, mean $\Delta T_1$ of 0.011 s; green line, median $\Delta T_1$ of 0.002 s). **F**: Overlaid KDEs of pre-contrast $T_1$ in NAWM. **G**: Overlaid KDE of $\Delta T_1$ in cGM (blue line, mean $\Delta T_1$ of -0.107 s; green line, median $\Delta T_1$ of -0.091 s). **H**: Per voxel Spearman's rho coefficient in cGM also shows a stronger correlation than NEL. **I**: Overlaid KDEs of pre-contrast $T_1$ in cGM. NEL = non-enhancing lesion, NAWM = normal-appearing white matter, cGM = cortical gray matter.

**Table 3. Correlation of $\Delta T_1$ with WML volume and count.**

| Tissue | $T_1$ metric (per subject) | WML volume | | | WML count | | | WML volume/count | | |
|---|---|---|---|---|---|---|---|---|---|---|
| | | r | p | adjusted p | r | p | adjusted p | r | p | adjusted p |
| WML | mean $\Delta T_1$ | 0.194 | 0.191 | 0.372 | -0.181 | 0.223 | 0.372 | 0.264 | 0.073 | 0.230 |
| | variance $\Delta T_1$ | 0.182 | 0.220 | 0.372 | **0.590** | **< 0.001** | **< 0.001** | -0.150 | 0.315 | 0.429 |
| | median $\Delta T_1$ | 0.160 | 0.284 | 0.426 | -0.272 | 0.064 | 0.230 | 0.261 | 0.077 | 0.230 |
| | IQR $\Delta T_1$ | 0.221 | 0.135 | 0.338 | **0.493** | **< 0.001** | **0.003** | -0.081 | 0.589 | 0.680 |
| | kurtosis $\Delta T_1$ | 0.106 | 0.479 | 0.599 | -0.010 | 0.949 | 0.949 | 0.042 | 0.777 | 0.833 |
| NEL | mean $\Delta T_1$ | 0.218 | 0.140 | 0.312 | -0.175 | 0.240 | 0.404 | 0.273 | 0.063 | 0.226 |
| | variance $\Delta T_1$ | 0.139 | 0.353 | 0.481 | **0.582** | **< 0.001** | **< 0.001** | -0.170 | 0.254 | 0.404 |
| | median $\Delta T_1$ | 0.164 | 0.269 | 0.404 | -0.270 | 0.066 | 0.226 | 0.262 | 0.075 | 0.226 |
| | IQR $\Delta T_1$ | 0.216 | 0.146 | 0.312 | **0.489** | **< 0.001** | **0.004** | -0.082 | 0.583 | 0.673 |
| | kurtosis $\Delta T_1$ | 0.005 | 0.975 | 0.975 | -0.126 | 0.400 | 0.500 | 0.055 | 0.712 | 0.763 |
| NAWM | mean $\Delta T_1$ | 0.175 | 0.240 | 0.450 | 0.203 | 0.172 | 0.392 | 0.011 | 0.941 | 0.998 |
| | variance $\Delta T_1$ | 0.244 | 0.099 | 0.392 | 0.341 | 0.019 | 0.286 | 0.000 | 0.998 | 0.998 |
| | median $\Delta T_1$ | 0.210 | 0.156 | 0.392 | 0.231 | 0.118 | 0.392 | 0.002 | 0.991 | 0.998 |
| | IQR $\Delta T_1$ | 0.198 | 0.183 | 0.392 | 0.235 | 0.111 | 0.392 | 0.028 | 0.851 | 0.998 |
| | kurtosis $\Delta T_1$ | -0.154 | 0.300 | 0.500 | -0.141 | 0.343 | 0.514 | -0.073 | 0.626 | 0.853 |
| cGM | mean $\Delta T_1$ | **0.397** | **0.006** | **0.043** | 0.339 | 0.020 | 0.074 | 0.159 | 0.287 | 0.391 |
| | variance $\Delta T_1$ | 0.261 | 0.076 | 0.163 | **0.423** | **0.003** | **0.043** | -0.069 | 0.647 | 0.746 |
| | median $\Delta T_1$ | **0.309** | **0.035** | 0.104 | 0.229 | 0.122 | 0.229 | 0.142 | 0.340 | 0.425 |
| | IQR $\Delta T_1$ | 0.263 | 0.074 | 0.163 | **0.356** | **0.014** | 0.070 | -0.031 | 0.838 | 0.838 |
| | kurtosis $\Delta T_1$ | -0.173 | 0.244 | 0.387 | -0.168 | 0.258 | 0.387 | 0.047 | 0.755 | 0.809 |

WML = white matter lesion; NEL = non-enhancing lesion; NAWM = normal-appearing white matter; cGM = cortical gray matter.

All p-values are given with actual numbers except values < 0.001. Coefficients with p-values less than 0.05 are shown in bold face. *Partial correlations after controlling for age, sex, and symptom duration are provided in* S1 Table.

EDSS (rho = -0.365, P = 0.01) scores. The interquartile range (IQR), variance, and kurtosis of $\Delta T_1$ values in NELs also all correlated with EDSS scores (Table 4). These measures indicate that wider spread (larger IQR, variance and smaller kurtosis) in NEL $\Delta T_1$ values, likely related to regions of larger lesion $\Delta T_1$ in the background of overall small $\Delta T_1$ are seen in cases with more advanced disability. Similar associations between variability in NEL $\Delta T_1$ and disability were seen for non-dominant hand 9HPT and SDMT scores (Tables 4 and 5) When adjusted for covariates of age, sex, and symptom duration, most of these correlations remained significant (S2 Table). However, adjustment of p-values for multiple comparisons resulted in loss of significance for all these relationships. IQR and kurtosis of $\Delta T_1$ in NAWM showed bivariate correlation with EDSS scores. The IQR of $\Delta T_1$ in cGM correlated with EDSS, 9HPTDOM, and 25FTW. However, nearly all the correlations noted between NAWM and cGM values and disability scores were no longer significant after adjustment for covariates and correction for multiple comparisons (S2 Table).

Median EDSS was 3.0 (range 1.0–6.5). $\Delta T_1$ in NELs was greater (P<0.0001) in participants with whose EDSS score was greater than the median (-0.168, range: -0.272 to 0.050) than those with EDSS scores at or below the median (-0.109, range: -0.281 to 0.134) (Fig 4A and Table 6). A wider IQR and lower positive kurtosis (thus, a wide-tailed distribution) for $\Delta T_1$ in NELs was also seen in those with EDSS scores above the median (Table 6). Gadolinium also induced greater changes in mean and median $T_1$ relaxation time of NELs in participants with progressive MS (SPMS and PPMS) phenotypes (median $\Delta T_1$: -0.177, range: -0.272 to -0.118) than those with RRMS (median $\Delta T_1$: -0.110, range: -0.281 to 0.134; P<0.001) (Fig 4B and Table 6).

**Table 4.** Correlation of per subject $\Delta T_1$ with clinical measures (Part I).

| Tissue | $T_1$ metric (per subject) | EDSS | | | SDMT | | | PASAT | | | MFIS | | |
|---|---|---|---|---|---|---|---|---|---|---|---|---|---|
| | | rho | p | adjusted p | r | p | adjusted p | r | p | adjusted p | r | p | adjusted p |
| WML | mean $\Delta T_1$ | -0.231 | 0.118 | 0.393 | -0.098 | 0.514 | 0.719 | 0.204 | 0.173 | 0.463 | -0.174 | 0.249 | 0.518 |
| | variance $\Delta T_1$ | 0.286 | 0.051 | 0.257 | -0.115 | 0.441 | 0.648 | 0.031 | 0.836 | 0.910 | 0.230 | 0.124 | 0.393 |
| | median $\Delta T_1$ | **-0.366** | **0.011** | 0.081 | 0.018 | 0.905 | 0.931 | 0.220 | 0.141 | 0.411 | -0.116 | 0.444 | 0.648 |
| | IQR $\Delta T_1$ | **0.365** | **0.012** | 0.081 | -0.312 | 0.033 | 0.192 | -0.030 | 0.841 | 0.910 | 0.172 | 0.252 | 0.518 |
| | kurtosis $\Delta T_1$ | **-0.388** | **0.007** | 0.081 | **0.384** | **0.008** | 0.081 | 0.049 | 0.745 | 0.899 | -0.002 | 0.989 | 0.989 |
| NEL | mean $\Delta T_1$ | -0.231 | 0.118 | 0.365 | -0.059 | 0.692 | 0.824 | 0.214 | 0.153 | 0.383 | -0.134 | 0.375 | 0.597 |
| | variance $\Delta T_1$ | **0.324** | **0.027** | 0.153 | -0.186 | 0.211 | 0.485 | 0.009 | 0.952 | 0.952 | 0.156 | 0.301 | 0.527 |
| | median $\Delta T_1$ | **-0.365** | **0.012** | 0.081 | 0.024 | 0.875 | 0.901 | 0.221 | 0.140 | 0.377 | -0.109 | 0.472 | 0.672 |
| | IQR $\Delta T_1$ | **0.367** | **0.011** | 0.081 | **-0.316** | **0.031** | 0.153 | -0.032 | 0.834 | 0.885 | 0.165 | 0.272 | 0.501 |
| | kurtosis $\Delta T_1$ | **-0.387** | **0.007** | 0.081 | **0.411** | **0.004** | 0.081 | 0.048 | 0.752 | 0.824 | -0.269 | 0.070 | 0.273 |
| NAWM | mean $\Delta T_1$ | 0.088 | 0.558 | 0.906 | 0.018 | 0.903 | 0.929 | 0.185 | 0.218 | 0.695 | -0.002 | 0.990 | 0.990 |
| | variance $\Delta T_1$ | 0.280 | 0.057 | 0.495 | -0.209 | 0.158 | 0.695 | -0.025 | 0.870 | 0.923 | 0.088 | 0.559 | 0.906 |
| | median $\Delta T_1$ | 0.047 | 0.753 | 0.906 | 0.036 | 0.810 | 0.906 | 0.188 | 0.210 | 0.695 | 0.033 | 0.829 | 0.906 |
| | IQR $\Delta T_1$ | **0.292** | **0.046** | 0.495 | -0.232 | 0.116 | 0.678 | -0.056 | 0.711 | 0.906 | 0.124 | 0.413 | 0.808 |
| | kurtosis $\Delta T_1$ | **-0.290** | **0.048** | 0.495 | 0.152 | 0.308 | 0.771 | 0.143 | 0.343 | 0.801 | -0.284 | 0.055 | 0.495 |
| cGM | mean $\Delta T_1$ | 0.020 | 0.896 | 0.896 | -0.047 | 0.754 | 0.896 | 0.252 | 0.091 | 0.366 | 0.024 | 0.876 | 0.896 |
| | variance $\Delta T_1$ | 0.252 | 0.087 | 0.366 | -0.163 | 0.272 | 0.502 | -0.119 | 0.431 | 0.580 | 0.100 | 0.507 | 0.657 |
| | median $\Delta T_1$ | -0.050 | 0.739 | 0.896 | 0.027 | 0.859 | 0.896 | 0.263 | 0.078 | 0.366 | -0.027 | 0.857 | 0.896 |
| | IQR $\Delta T_1$ | 0.278 | 0.058 | 0.366 | -0.224 | 0.131 | 0.382 | -0.119 | 0.430 | 0.580 | 0.167 | 0.269 | 0.502 |
| | kurtosis $\Delta T_1$ | -0.243 | 0.100 | 0.366 | 0.244 | 0.098 | 0.366 | 0.166 | 0.271 | 0.502 | -0.166 | 0.270 | 0.502 |

WML = white matter lesion; NEL = non-enhancing lesion; NAWM = normal-appearing white matter; cGM = cortical gray matter; IQR = inter-quartile range;
rho = Spearman's rho coefficient; r = Pearson's correlation coefficient; p = p-values; adjusted p = p-value corrected for multiple comparison (false discovery rate: FDR);
EDSS = Expanded Disability Status Scale; SDMT = Symbol Digit Modalities Test; PASAT = Paced Auditory Serial Addition Test; MFIS = Modified Fatigue Impact Scale
All p-values are given with actual numbers except values < 0.001. Coefficients with p-values less than 0.05 are shown in bold face. *Partial correlations after controlling for age, sex, and symptom duration are provided in* S2 Table.

The variance and IQR were also elevated in those with progressive phenotypes (Table 6). Mean and median $\Delta T_1$ in NELs were greater in subjects not on MS treatment (Fig 4C and Table 6). In further analysis of treatment effect using logistic regression, $\Delta T_1$ significantly predicted whether a participant was on treatment (P = 0.01), even when adjusted for MS phenotype (P = 0.045) (Table 7).

## Discussion

This study demonstrates that a shift in $T_1$ relaxation time caused by gadolinium can be measured in WMLs in MS by MP2RAGE on 7T MRI, even when visible contrast enhancement cannot be observed. Further, the magnitude and variability of this $T_1$-shift (termed $\Delta T_1$ in this study) is associated with the severity of MS-related disability and MS-phenotype and $\Delta T_1$ magnitude is associated with use of immunomodulatory MS treatment. These findings have potential implications towards measurement of disease activity and treatment effect monitoring in MS.

In order to place these findings into proper context, the underlying mechanism leading to an appreciable $\Delta T_1$ in lesions warrants further exploration. Gadolinium-based contrast agents (GBCAs) are utilized in MR imaging because of their extracellular distribution and reduced $T_1$ relaxation rates [26]. In a healthy central nervous system, gadolinium does not leak into the

**Table 5. Correlation of per subject $\Delta T_1$ with clinical measures (Part II).**

| Tissue | $T_1$ metric (per subject) | 9HPTDOM | | | 9HPTNONDOM | | | 25FTW | | |
|---|---|---|---|---|---|---|---|---|---|---|
| | | r | p | adjusted p | r | p | adjusted p | r | p | adjusted p |
| WML | mean $\Delta T_1$ | 0.147 | 0.329 | 0.564 | -0.034 | 0.820 | 0.910 | -0.086 | 0.567 | 0.764 |
| | variance $\Delta T_1$ | 0.266 | 0.074 | 0.322 | 0.027 | 0.858 | 0.910 | 0.143 | 0.338 | 0.564 |
| | median $\Delta T_1$ | 0.051 | 0.737 | 0.899 | -0.073 | 0.628 | 0.814 | -0.164 | 0.271 | 0.520 |
| | IQR $\Delta T_1$ | **0.398** | **0.006** | 0.081 | 0.131 | 0.382 | 0.607 | 0.255 | 0.084 | 0.325 |
| | kurtosis $\Delta T_1$ | -0.199 | 0.185 | 0.463 | -0.160 | 0.282 | 0.520 | -0.190 | 0.201 | 0.469 |
| NEL | mean $\Delta T_1$ | 0.128 | 0.397 | 0.604 | -0.055 | 0.713 | 0.824 | -0.099 | 0.508 | 0.684 |
| | variance $\Delta T_1$ | **0.301** | **0.042** | 0.184 | 0.063 | 0.673 | 0.824 | 0.166 | 0.266 | 0.501 |
| | median $\Delta T_1$ | 0.048 | 0.754 | 0.824 | -0.076 | 0.613 | 0.794 | -0.166 | 0.266 | 0.501 |
| | IQR $\Delta T_1$ | **0.399** | **0.006** | 0.081 | 0.133 | 0.372 | 0.597 | 0.256 | 0.083 | 0.290 |
| | kurtosis $\Delta T_1$ | -0.184 | 0.222 | 0.485 | -0.106 | 0.480 | 0.672 | -0.227 | 0.125 | 0.365 |
| NAWM | mean $\Delta T_1$ | 0.118 | 0.435 | 0.808 | -0.034 | 0.822 | 0.906 | -0.041 | 0.782 | 0.906 |
| | variance $\Delta T_1$ | 0.238 | 0.112 | 0.678 | 0.054 | 0.718 | 0.906 | 0.118 | 0.428 | 0.808 |
| | median $\Delta T_1$ | 0.117 | 0.439 | 0.808 | -0.034 | 0.819 | 0.906 | -0.041 | 0.785 | 0.906 |
| | IQR $\Delta T_1$ | 0.196 | 0.191 | 0.695 | 0.056 | 0.711 | 0.906 | 0.174 | 0.242 | 0.705 |
| | kurtosis $\Delta T_1$ | -0.160 | 0.288 | 0.771 | -0.061 | 0.682 | 0.906 | -0.197 | 0.184 | 0.695 |
| cGM | mean $\Delta T_1$ | 0.032 | 0.831 | 0.896 | -0.180 | 0.226 | 0.502 | -0.134 | 0.370 | 0.577 |
| | variance $\Delta T_1$ | 0.265 | 0.075 | 0.366 | 0.131 | 0.379 | 0.577 | 0.198 | 0.183 | 0.492 |
| | median $\Delta T_1$ | 0.030 | 0.842 | 0.896 | -0.142 | 0.340 | 0.577 | -0.185 | 0.212 | 0.502 |
| | IQR $\Delta T_1$ | 0.273 | 0.066 | 0.366 | 0.118 | 0.429 | 0.580 | 0.240 | 0.105 | 0.366 |
| | kurtosis $\Delta T_1$ | -0.235 | 0.116 | 0.370 | -0.140 | 0.348 | 0.577 | -0.245 | 0.097 | 0.366 |

WML = white matter lesion; NEL = non-enhancing lesion; NAWM = normal-appearing white matter; cGM = cortical gray matter; IQR = inter-quartile range;

rho = Spearman's rho coefficient; r = Pearson's correlation coefficient; p = p-values; adjusted p = p-value corrected for multiple comparison (false discovery rate: FDR);

9HPTDOM = 9-hole peg test for dominant hand; 9HPTNONDOM = 9-hole peg test for non-dominant hand; 25FTW = timed 25-foot walk.

All p-values are given with actual numbers except values < 0.001. Coefficients with p-values less than 0.05 are shown in bold face. *Partial correlations after controlling for age, sex, and symptom duration are provided in* S2 Table.

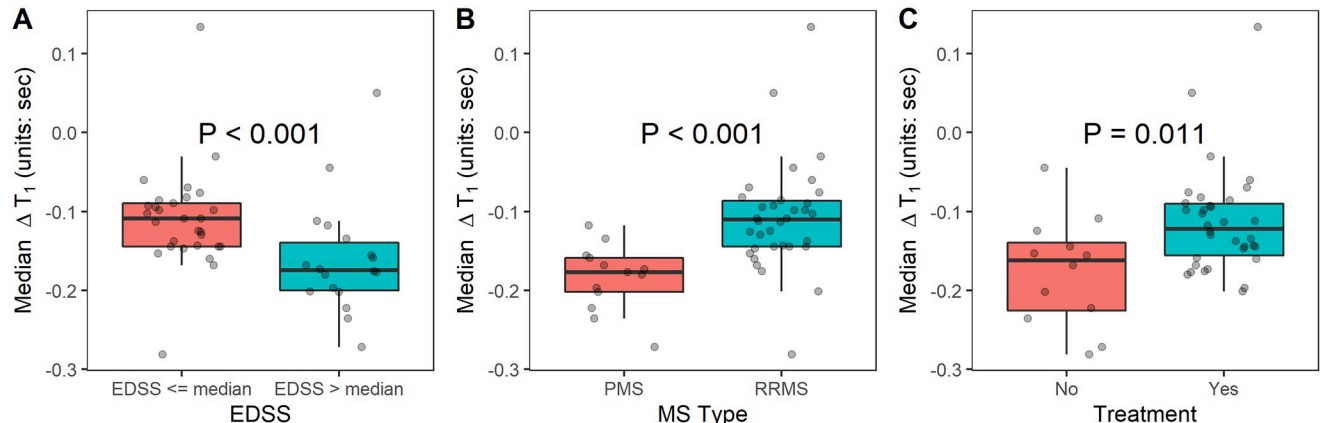

**Fig 4. Group differences in median $\Delta T_1$ in non-enhancing white matter lesions.** A: NEL median $\Delta T_1$ comparison for EDSS scores that were > median (3.0) vs. ≤ median. Gadolinium induced greater changes in $T_1$ in MS participants with higher EDSS scores. B: NEL median $\Delta T_1$ comparison for MS Type, PMS vs. RRMS. Gadolinium induced greater changes in $T_1$ in participants with PMS. C: NEL median $\Delta T_1$ comparison for multiple sclerosis immunomodulatory treatment, Yes vs. No. Gadolinium induced greater changes in $T_1$ in participants not on MS treatment. Per subject median $\Delta T_1$ values were used in the Mann-Whitney U-test. NEL = non-enhancing lesion; EDSS = Expanded Disability Status Scale; MS = multiple sclerosis; PMS = progressive multiple sclerosis (PPMS + SPMS); PPMS = primary progressive multiple sclerosis; SPMS = secondary progressive multiple sclerosis; RRMS = relapsing-remitting multiple sclerosis.

**Table 6. Group differences in $\Delta T_1$ metrics.**

| NEL $T_1$ metric | EDSS | | MS type | | Treatment | |
|---|---|---|---|---|---|---|
| | ≤ Median Score | > Median Score | PMS | RRMS | No | Yes |
| mean $\Delta T_1$ | -0.120 | **-0.186** * | -0.187 | **-0.122** ** | -0.192 | **-0.126** ** |
| | (-0.304, 0.174) | (-0.238, 0.100) | (-0.238, -0.062) | (-0.304, 0.174) | (-0.304, -0.049) | (-0.231, 0.174) |
| median $\Delta T_1$ | -0.109 | **-0.168** *** | -0.177 | **-0.110** *** | -0.162 | **-0.122** * |
| | (-0.281, 0.134) | (-0.272, 0.050) | (-0.272, -0.118) | (-0.281, 0.134) | (-0.281, -0.045) | (-0.201, 0.134) |
| variance $\Delta T_1$ | 0.081 | **0.118** *** | 0.119 | **0.082** ** | 0.111 | 0.083 |
| | (0.026, 0.297) | (0.006, 0.366) | (0.048, 0.366) | (0.006, 0.297) | (0.006, 0.366) | (0.040, 0.337) |
| IQR $\Delta T_1$ | 0.272 | **0.382** ** | 0.357 | **0.272** * | 0.337 | 0.297 |
| | (0.142, 0.439) | (0.118, 0.702) | (0.229, 0.702) | (0.118, 0.469) | (0.118, 0.702) | (0.193, 0.689) |
| kurtosis $\Delta T_1$ | 6.425 | **5.102** ** | 5.255 | 6.205 | 5.837 | 6.023 |
| | (4.459, 14.589) | (2.636, 8.803) | (3.843, 8.803) | (2.636, 14.589) | (2.636, 14.589) | (3.843, 11.825) |

Inter-subject median (range) is shown for each metric. Significance for group difference tested by Mann-Whitney U-test shown. Statistically significances (*: $P < 0.05$, **: $P < 0.01$, ***: $P < 0.001$) are shown in bold.

NEL = non-enhancing lesion; MS = multiple sclerosis; RRMS = relapsing-remitting multiple sclerosis; PMS = progressive MS (primary progressive + secondary progressive multiple sclerosis); IQR = inter-quartile range

extravascular space or accumulate in tissues due to the actions of the BBB. Focal disruption of the BBB leads to gadolinium accumulation, which is an established marker of acute inflammation in MS lesions [27]. However, the process of BBB disruption in MS is not exclusive to acutely inflamed WMLs. The BBB, which is composed of a glycolax layer, non-fenestrated endothelium with tight junctions, basal lamina, and astrocytic end feet, may be breached through both disruptive and non-disruptive processes [28]. Transient inflammatory changes are typically non-disruptive, resulting in increases in the concentration and activity of cellular adhesion molecules and transporters, along with changes to astrocyte activity [26]. More profound BBB injury can result in disruptive changes to the BBB, including degeneration of the glycolax layer, damage to endothelium and glia limitans, breakdown of tight junctions, and astrocytopathy [28]. Histopathologic examination of MS lesions confirms BBB disruption in acutely inflamed MS lesions, including basal lamina degradation and deposition of vascular fibrinogen within lesions [29, 30]. These findings are not exclusive to acutely inflamed lesions, however, as chronic-active lesions also show fibrinogen deposition, in addition to deposition of IgG, endothelial cell changes, and leukocyte activity in widened Virchow-Robin spaces [29,

**Table 7. Multiple logistic regression analysis of significance of treatment effect on $\Delta T_1$.**

| Model | $\beta_0$ | $\beta_1$ | $\beta_2$ |
|---|---|---|---|
| | p-value | p-value | p-value |
| $Tx = \beta_0 + \beta_1 \times \Delta T_1$ | 3.846** | 19.176* | N/A |
| | P = 0.002 | P = 0.013 | N/A |
| $Tx = \beta_0 + \beta_1 \times \Delta T_1 + \beta_2 \times (MS\ type)$ | 3.869** | 17.967* | -0.129 |
| | P = 0.002 | P = 0.045 | P = 0.80 |

Simple logistic regression model in first row, followed by multiple logistic regression including MS Type (relapsing-remitting versus progressive). Tx = MS treatment status (1 = yes, 0 = no); MS = multiple sclerosis, N/A = not applicable;

*: $P < 0.05$;

**: $P < 0.01$.

31]. To a lesser degree, chronic-inactive lesions also show similar signs of BBB breakdown on histopathology [29].

Given evidence of BBB breakdown in WMLs of all types, extravasation of gadolinium into lesioned tissue is likely the mechanism by which $\Delta T_1$ changes occurred in this study. Although an increase in blood flow could potentially cause a $\Delta T_1$ shift through increased presence of intravascular gadolinium, all histopathologic and MRI evidence points towards reduced vascularity and blood flow in the white matter and WMLs of patients with MS [32]. Further, the mean distribution half-life of most GBCAs are approximately 10 minutes, and thus by the time of acquisition of MP2RAGE images in this study the plasma concentration of gadolinium has markedly decreased, whereas tissue concentrations are peaked [33].

Measurement of $\Delta T_1$ in NELs in this study is thus revealing of BBB breakdown in WMLs without visible contrast enhancement. The minimal $\Delta T_1$ in NAWM shows that this is specific to lesions and correlations with disability and phenotype confirm clinical relevancy. These findings are in line with prior data performing measurements of BBB integrity in MS lesions by other neuroimaging methodologies. Previous work using $T_1$ maps calculated from combination of a $T_1$-weighted spin-echo sequence and a PD-weighted gradient-echo sequence before and after triple-dose gadolinium (0.3 mmol/kg) consistently showed a $T_1$ shift with contrast in NELs in MS, which was not apparent in NAWM [10, 34]. The difference between NELs and NAWM was not profound at 5 minutes after contrast injection, but became more pronounced later and was sustained at time points as far out as 60 minutes. Our study did not include multiple post-gadolinium acquisition time points, so such a comparison cannot be made. However, although these prior studies had similar findings, $T_1$-shortening in NELs was not related to disability levels or phenotype, and piloting of this technique in a placebo-controlled trial of natalizumab showed no treatment effect [35]. DCE MRI is also capable of measuring BBB disruption in NELs, with findings of increased permeability in NELs in MS patients with recent relapses and decreased permeability in those on disease modifying drugs [8]. Similar findings were seen in this study, with participants on MS treatment having a smaller $\Delta T_1$ in NELs. The significance of this finding is unclear. While it is possible this was influenced by a bias towards refraining from disease modifying therapy in very advanced, especially progressive MS phenotypes, adjustment for MS phenotype did not alter the relationship. Given that the treatment effect of some MS therapies, such as interferon-beta, is partially mediated by improvements in BBB function [36], it is quite possible that the measurement of $\Delta T_1$ in this study revealed less BBB breakdown as a function of immunomodulatory therapy. Such findings indicate the potential applicability of this technique as a clinical trial outcome measure.

Although similar to prior work, our proposed concept of MP2RAGE $\Delta T_1$ mapping as a means by which to study BBB integrity in MS stands out because of potential greater applicability of the technique to clinical trials and clinical practice than other methods. Unlike some prior methodologies, MP2RAGE $\Delta T_1$ mapping does not require triple-dose gadolinium. Widespread use of triple-dose gadolinium is impractical given safety concerns such as allergic reactions, nephrogenic systemic sclerosis, and recent evidence showing deposition of gadolinium in the dentate nucleus and globus pallidus [37]. Further, unlike other methods, creation of $T_1$ maps from MP2RAGE does not require multiple image acquisitions with later synthesis. This property allows rapid and robust production of $T_1$ maps, which can readily be created on-scanner with automated, manufacturer-provided tools on some scanning platforms. Analysis is also simple, with direct subtraction of $T_1$ maps and interpretation of the absolute change after gadolinium administration. This is in direct contrast to the complex mathematical modeling required to derive and interpret permeability measurements from DCE MRI [38]. MP2RAGE $T_1$ maps are also likely more reliable than other $T_1$ mapping techniques, as the inhomogeneity correction inherent to processing of MP2RAGE images removes the influence

of scanner field inhomogeneity and the acquisition technique minimizes the influence of $T_2^*$ and proton density effects, resulting in a more 'pure' $T_1$ map [11]. It is likely that these properties, along with the higher resolution affordable at 7T, allowed us to quantify $\Delta T_1$ in NELs in a manner that was more-clinically relevant than prior work. Although 7T was used in this study, it should be noted that the MP2RAGE technique is not exclusive to 7T and can easily be performed and processed on clinical scanners. Further, given that the r1 relaxivity of some GBCAs is reduced at higher magnetic fields [39], it is possible that similar, or possibly more robust findings can be seen at 1.5T or 3T. This effect may be counter-balanced, however, by the greater separation of $T_1$ relaxation values between tissue types and increased resolution at 7T [11, 14]. Given potential equipoise regarding field strength, future attempts at replication of our findings should aim to perform direct comparisons of contrast-enhanced MP2RAGE at 3T and 7T. An inability to confirm our findings on 3T would clearly limit potential clinical applications and use in clinical trials.

The need for alternate means by which to subclassify MS lesions is clear, as imaging-pathology correlations reveal that differentiation of active, chronic-active, and chronic-inactive lesions by $T_2$ signal characteristics alone is not possible [29]. Although visible contrast enhancement differentiates active from chronic lesions, differentiation of chronic lesions into those that have continued inflammation versus those that do not is more difficult. Multiple putative imaging measures for such chronic inflammatory lesion changes have been proposed, including slowly expanding lesions and lesions with paramagnetic rims [40]. If chronic BBB breakdown is indicative of chronic inflammation, $\Delta T_1$ measurement may be a means by which to quantify chronic inflammatory changes in MS lesions and differentiation of such lesions from those that are chronic-inactive. This hypothesis could potentially be studied in the future by applying the $\Delta T_1$ technique in a multi-modal study including susceptibility-weighted images, with comparisons between lesions with and without paramagnetic rims. Establishing a unique link between $\Delta T_1$ and paramagnetic rims. Establishing a unique link between $\Delta T_1$ and paramagnetic rims (as a surrogate for chronic-active inflammation could potentially provide additional tools for) quantification of disease activity in SPMS and PPMS, in whom contrast enhancing lesions are rarely seen and chronic-active lesions are more pathologically predominant [41, 42]. Despite the rarity of visually enhancing lesions in progressive phenotypes, BBB breakdown clearly occurs, as studies of triple-dose gadolinium and delayed acquisition of images after contrast reveal significant increases in the number of enhancing lesions visible in PPMS [41]. Given our findings of a more profound and variable $\Delta T_1$ in NELs seen in those with progressive MS, we propose further evaluation of this technique as a means by which to monitor progression.

It is unclear if the $\Delta T_1$ changes seen here are due to non-disruptive BBB changes, perhaps occurring in the setting of increased cytokine release and microglial activity due to chronic inflammation, versus more permanent disruptive changes to the BBB that occurred at lesion formation. Differentiation between these two processes could be attempted by comparing lesions with and without paramagnetic rims, but ultimate proof will likely require *in vivo* imaging followed by post-mortem histopathology. However, given that prolonged $T_1$ relaxation times (seen as darker 'black holes' on $T_1$-w images) strongly correlate with the degree of tissue damage within WMLs [43], our results showing only a weak correlation between pre-contrast $T_1$ and $\Delta T_1$ values suggest the severity of a lesion is not the sole determinant of BBB leak. Further, the correlation between pre-contrast $T_1$ and $\Delta T_1$ values in NELs was weaker than the same correlation evaluated in NAWM or cGM—further supporting $\Delta T_1$ as a metric with more specificity to BBB disruption as a function of lesional-pathology rather than widespread tissue alterations. These data together suggest that lesion-based $\Delta T_1$ provides additional information beyond $T_1$ alone, warranting further investigation as an MS outcome measure.

In addition to a lack of histopathologic correlations, there are a few other potential limitations to consider when evaluating the conclusions of this work. The lack of healthy control participants, a limitation in this study, precluded comparison of NAWM of MS brains and healthy brains. Including healthy participants in the future study would help with the interpretation of the patient data. Although our sample size is relatively large in the context of 7T research in MS, larger sample sizes will be needed to confirm these findings and evaluate widespread applicability. It is quite likely that the small sample size of the study is responsible for reductions in the number of significant correlations with correction for multiple comparisons and/or with adjustment for co-variates. Greater statistical power is necessary to determine if the relationships seen in univariate analysis are due to false discovery or covariance with other factors. We hope that the initial findings of this work are hypothesis stimulating and that our group and others can work to either refute or confirm the findings with larger, future work.

This study is also limited by its cross-sectional nature, and determination as to whether these findings track with disability progression or are modified by treatment will likely require longitudinal analyses. Additionally, as lesion masks were drawn as one volume for each subject, our study does not permit the specificity of lesion-by-lesion analysis within subjects beyond visual inspection of heat maps. Individual lesion analyses are beyond the scope and capability of this dataset but could be evaluated in the future. However, individual lesion analyses are potentially fraught with limitations of their own, such as unclear lesion borders in regions of large, confluent periventricular demyelination. An additional limitation of our paper due to the lack of lesion-by-lesion masks is the inability to separate out the influence of any non-demyelinating (i.e. microvascular ischemic) lesions on our results. We expect that any such contribution would be extremely small, but cannot rule out this possibility, especially in older subjects in the cohort. Finally, although the MP2RAGE technique employed does help to overcome receive field (B$_1^-$) inhomogeneity, it has been noted that the accuracy of T$_1$ estimation can also be influenced by transmit field (B$_1^+$) inhomogeneity. Such inhomogeneities are especially relevant for T$_1$ estimation at ultra-high field. The influence of this homogeneity, along with rare voxel misalignments in co-registration likely explain the small proportion of positive ΔT$_1$ voxels seen in this work, which is similar to prior evaluations of T$_1$ change with gadolinium [34]. To overcome field-induced limitations in T$_1$ mapping, more recent work has evaluated measurement of B$_1^+$ field maps by techniques such as the Sa2RAGE sequence, with integration of these maps into T$_1$ map calculations [44]. Despite our lack of B$_1^+$ field map acquisition significant and clinically relevant findings were obtained by our protocol. Thus, future work including Sa2RAGE or other similar methods would only likely improve upon the ability to measure ΔT$_1$ in NELs in MS.

## Conclusion

Despite limitations, the findings of this study lead to a conclusion that T$_1$ mapping by MP2RAGE shows promise as a novel means by which to assess MS disease severity and potentially to monitor treatment effect. Integration of clinically feasible evaluations of BBB integrity in MS will likely help give greater insight into MS pathology, and may provide an opportunity to evaluate chronic lesion inflammation.

## Supporting information

**S1 Table. Partial correlation of ΔT$_1$ with WML volume and count.**
(DOCX)

**S2 Table. Partial correlation of per subject $\Delta T_1$ with clinical measures.**
(DOCX)

## Acknowledgments

The investigators are grateful to research nurses Kerry Naunton and Julie Fiol and MRI technicians Terri Brawner, Kathleen Kahl, and Ivana Kusevic—all of whom were critical to implementation of the study.

## Author Contributions

**Conceptualization:** Seongjin Choi, Margaret Spini, Daniel M. Harrison.

**Data curation:** Seongjin Choi, Daniel M. Harrison.

**Formal analysis:** Seongjin Choi, Daniel M. Harrison.

**Funding acquisition:** Daniel M. Harrison.

**Investigation:** Seongjin Choi, Daniel M. Harrison.

**Methodology:** Seongjin Choi, Margaret Spini, Daniel M. Harrison.

**Project administration:** Daniel M. Harrison.

**Resources:** Jun Hua, Daniel M. Harrison.

**Software:** Seongjin Choi, Daniel M. Harrison.

**Supervision:** Daniel M. Harrison.

**Validation:** Seongjin Choi, Daniel M. Harrison.

**Visualization:** Seongjin Choi, Daniel M. Harrison.

**Writing – original draft:** Seongjin Choi, Daniel M. Harrison.

**Writing – review & editing:** Seongjin Choi, Margaret Spini, Jun Hua, Daniel M. Harrison.

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
