## [Decision Letter · Decision Letter 0]

19 Nov 2020

PONE-D-20-33106

Blood-brain barrier breakdown in non-enhancing multiple sclerosis lesions detected by 7-Tesla MP2RAGE ΔT1 mapping

PLOS ONE

Dear Dr. Harrison,

Thank you for submitting your manuscript to PLOS ONE. After careful consideration, we feel that it has merit but does not fully meet PLOS ONE’s publication criteria as it currently stands. Therefore, we invite you to submit a revised version of the manuscript that addresses the points raised during the review process.

In addition to the comments raised by the Reviewers, please also address the following comments:

1. The word “gender” is used throughout the manuscript. Usually, “sex” (the biological designation) is meant. “Gender” is the social construct and is rarely relevant in neurologic disease. Please revise the text to use “sex” rather than “gender” throughout.

2. The "Symbol Digit Modalities Test" is incorrectly referred to as the "symbol digits modalities test" (note the lack of pluralization of digits and capitalization in the correct spelling).

3. The point about potentially inhibiting hypothesis-generating research due to multiple comparisons is understood, some correction is warranted in the context of your manuscript. Table 3 and Table 4 have 45 and 105 tests, respectively. While Bonferroni correction would indeed be overly conservative, something like false discovery rate correction be considered.

4. P-values, by definition, cannot be equal to 0.0, as reported in Table 4. Please update accordingly.

5. While the comments about the benefit of the MP2RAGE sequence with respect to not requiring co-registration for T1 measurements is acknowledged, this point is somewhat weakened later on when you refer to the subtraction of T1 maps in the context of the current manuscript, which does require co-registration. Consider mentioning this.

We look forward to receiving your revised manuscript.

Kind regards,

Niels Bergsland

Academic Editor

PLOS ONE

Journal Requirements:

2. Please include your Ethics statement in your Methods section. Please ensure that this states the full name of the IRB and states that the IRB specifically approved the study. Please also provide the type of consent obtained.

*In your revised cover letter, please address the following prompts:

"I have read the journal's policy and the authors of this manuscript have the following competing interests: DMH has received consulting fees from EMD Serono, Inc., Genentech, Sanofi-Genzyme, and Biogen.".

 i) Please confirm that this does not alter your adherence to all PLOS ONE policies on sharing data and materials, by including the following statement: "This does not alter our adherence to  PLOS ONE policies on sharing data and materials.” (as detailed online in our guide for authors http://journals.plos.org/plosone/s/competing-interests).  If there are restrictions on sharing of data and/or materials, please state these. Please note that we cannot proceed with consideration of your article until this information has been declared.

ii) Please include your updated Competing Interests statement in your cover letter; we will change the online submission form on your behalf.

Reviewers' comments:

Reviewer's Responses to Questions

**Comments to the Author**

1. Is the manuscript technically sound, and do the data support the conclusions?

Reviewer #1: Partly

Reviewer #2: Partly

2. Has the statistical analysis been performed appropriately and rigorously? 

Reviewer #1: Yes

Reviewer #2: Yes

3. Have the authors made all data underlying the findings in their manuscript fully available?

Reviewer #1: Yes

Reviewer #2: No

4. Is the manuscript presented in an intelligible fashion and written in standard English?

Reviewer #1: Yes

Reviewer #2: Yes

5. Review Comments to the Author

Reviewer #1: Dr Choi and colleagues report a cross-sectional study to investigate if there is a BBB disruption in non-enhancing lesions (NEL) evaluating changes in T1 relaxation time in pre-and post-contrast MRI using MP2RAGE at 7T. They found that subtraction of pre-and post-contrast T1 maps, ΔT1, was greater in NEL than in NAWM and correlated with EDSS. While it is a novel approach, there are former articles at lower resolution MRI showing that NELs have increased permeability compared to NAWM.

I have major concerns.

• Abstract: The authors propose that NEL ΔT1 should be further evaluated as a biomarker for disease severity and treatment effect in MS. I would rather say that it should be evaluated as a possible biomarker of persistently disrupted BBB.

• Introduction: They state: -The existence of chronic-active lesions, along with the relative inability to differentiate these from inactive lesions on standard MRI sequences, demonstrates need additional in vivo markers for chronic inflammation in MS-. However, there are articles that showed in vivo and postmorten paramagnetic rims in susceptibility weighted imaging at 7, that correlated with chronic active lesions in pathological analysis. The presence of paramagnetic RIMS was also assessed at 3T MRI. A paragraph describing these results and data on slowly expanding lesions should be added.

• Materials and methods: Was MP2RAGE the only sequence acquired?

• Results: Data on the disease duration should be included. What percentage of progressive patients where under treatment? These data are important for the interpretation of the results.

• Discussion:

I. Paragraph-4 Authors claim that the magnitude and variability of ΔT1 is associated with the severity of MS-related disability, MS-phenotype and the use of immunomodulatory MS treatment. To better analyze the results, it is important to include disease duration and percentage of progressive MS under treatment.

II. Paragraph-5 As they found that ΔT1 revealed less BBB breakdown as a function of immunomodulatory therapy, they highlight the potential applicability of this technique as an outcome measure in clinical trials, and in clinical practice. The authors should deeply clarify that MP2RAGE at 7T is not currently feasible in clinical trials setting or in clinical practice. As they mention in the last sentence of that paragraph, their findings need to be replicated first at 3T.

III. Paragraph-7 They propose that changes in ΔT1 may be due to increased cytokine release and microglial activity due to chronic inflammation, or more permanent disruptive changes to the BBB that occurred at lesion formation. They claim that the differentiation between these two processes would likely require in vivo imaging followed by post-mortem histopathology. However, I think that a good approach to elucidate possible mechanisms involved would be to see if there is a gradient of ΔT1 in individual lesions, if a greater ΔT1 is detected at the edge of the lesion, this would be in favor of chronic inflammation in the border of the lesion. Furthermore, if susceptibility-weighted images show paramagnetic rims in NEL with greater ΔT1, it would suggest chronic inflammation in the edge of the lesion. Could the authors do a more detailed analysis?

• Conclusion: I would focus the conclusion on that using MP2RAGE at 7T, persistent disruption of the BBB was found. The last sentence mentions that there are currently no stablished biomarkers, I would remove it, there are promising biomarkers.

Reviewer #2: Thank you for this interesting study.

Please reformat Table 1 to show the variables subdivided by groups of MS category. It is important to know the relative ages and disability levels

The average age of this cohort was 46.2 (11.2). WM lesions of vascular origin may have also occurred in this cohort. Did you make an attempt to distinguish these from MS lesions?

Why did you chose EDSS = 5.0 as a threshold for binarization? This is not particularly clinically meaningful. Since 3.0 was the median in your cohort (and 6.5 the max), and since this is more meaningful (mild vs moderate disability), this might be a more appropriate threshold.

The phrase ‘inter-subject mean of per subject median’ is not well explained. Suggest rephrase or explain more fully in methods.

You found a pre/post-contrast difference in NAWM when using the median but not the mean – how do you explain this?

You report that delta T1 was significantly greater in NELs than in NAWM. But NELs also had a higher pre-contrast T1 than NAWM (as expected). Therefore, if both NELs and NAWM had the same permeability, and accumulated the same concentration of contrast, NELs would have a higher delta T1 simply because of the higher pre-contrast T1 (about double – though you find a bigger change than this). Presumably in controls you find a higher delta T1 in GM vs WM? You should report delta T1 for NAGM in this cohort and compare to NAWM. I am concerned that using delta T1 to make inferences about BBB permeability is difficult. You report a weak correlation between pre-contrast T1 and delta T1 – this deserves more discussion and a figure. Advise redo the correlation but within tissue types – i.e. within NAWM, GM, etc. If the correlation is weak in NELs, then this may be due to heterogeneity in some other factor – like permeability (or vascular surface area). This would fit with your other findings of lesional heterogeneity.

What was the rationale for a 10 minute gap between injection and post-contrast scan? Would a longer gap have influenced results?

I do not see that you acquired a T2-weighted or FLAIR sequence for lesion detection. If lesion masks were created using T1 information only, it is likely that some lesions were misclassified as NAWM, and that the lesions mask was biased towards black holes. You report the median T1 of lesions to be 2430 ms – this is very long and suggests that a significant number of black holes were present. The leakage behaviour of black holes is unlikely to be representative of T1-isointense lesions.

Has the raw data been provided? I cannot see in this manuscript. I assume this will follow in a supplementary material.

6. PLOS authors have the option to publish the peer review history of their article (what does this mean?). If published, this will include your full peer review and any attached files.

Reviewer #1: No

Reviewer #2: No

---

## [Author Response · Author response to Decision Letter 0]

9 Mar 2021

Please see the attached "Response to Reviewers" document attached to the submission. Full response could not be pasted here due to formatting, figures, etc.

---

## [Decision Letter · Decision Letter 1]

23 Mar 2021

PONE-D-20-33106R1

Blood-brain barrier breakdown in non-enhancing multiple sclerosis lesions detected by 7-Tesla MP2RAGE ΔT1 mapping

PLOS ONE

Dear Dr. Harrison,

Thank you for submitting your manuscript to PLOS ONE. After careful consideration, we feel that it has merit but does not fully meet PLOS ONE’s publication criteria as it currently stands. Therefore, we invite you to submit a revised version of the manuscript that addresses the points raised during the review process.

Please see below for one outstanding issue. I agree with the Reviewer that this point would be beneficial to discuss.

We look forward to receiving your revised manuscript.

Kind regards,

Niels Bergsland

Academic Editor

PLOS ONE

Journal Requirements:

Reviewers' comments:

Reviewer's Responses to Questions

**Comments to the Author**

1. If the authors have adequately addressed your comments raised in a previous round of review and you feel that this manuscript is now acceptable for publication, you may indicate that here to bypass the “Comments to the Author” section, enter your conflict of interest statement in the “Confidential to Editor” section, and submit your "Accept" recommendation.

Reviewer #1: (No Response)

Reviewer #2: All comments have been addressed

2. Is the manuscript technically sound, and do the data support the conclusions?

Reviewer #1: Yes

Reviewer #2: Yes

3. Has the statistical analysis been performed appropriately and rigorously? 

Reviewer #1: Yes

Reviewer #2: Yes

4. Have the authors made all data underlying the findings in their manuscript fully available?

Reviewer #1: Yes

Reviewer #2: Yes

5. Is the manuscript presented in an intelligible fashion and written in standard English?

Reviewer #1: Yes

Reviewer #2: Yes

6. Review Comments to the Author

Reviewer #1: Response to comment from previous round:

We thank you for the suggestion. Unfortunately, we suggest that such an analysis would be well beyond the scope of this paper and also may not provide any definitive conclusion. Regarding scope, as GRE images were not reviewed for paramagnetic rims and paramagnetic rim lesions were not separately masked, this would add multiple additional processing/identification/masking/registration/analysis steps – all of which would have to be performed, described, and evaluated in detail on their own in this already large paper (both in size and concept). As a first description of the ΔT1 concept requires much more detail on MP2RAGE ΔT1 methods and analysis, we feel there is not sufficient room to do so at this time. We do agree that, perhaps, in a future analysis, once the ΔT1 concept is described in this paper and this paper can be referred to, a multi-modal analysis of the type suggested could be done. Further, it should be noted that while lesions with paramagnetic rims do indeed correlate well with chronic-active lesions at autopsy, the finding of paramagnetic rims also has been described in acutely inflamed lesions, so specificity may complicate interpretation. Regarding the spatial analysis suggested – unfortunately that is not possible with the current WML masks created for this paper. As described in the methods, each individual WML was not mapped on its own. Rather, a full comprehensive WML mask was created for each individual subject. The full 3D volume of this mask and an enhancing WML mask for each subject were subtracted to create the full 3D volume for NELs for each subject. The analysis suggested would require each individual lesion in each individual subject to separated into individual volumes and then creation of a geometric methodology for separating the inner and outer portions of those volumes. While that could be the goal of a later undertaking and report, the idea for which we thank the reviewer, this would not be possible at this time with this dataset. We should note that our original Discussion section (last paragraph) acknowledges the limitation of the full volume approach, rather than a lesion-by-lesion analysis, but also discusses the biases/limitations of lesion-by-lesion analyses.

Although this does not entirely answer the reviewer’s request, though, it should be noted that a qualitative review of ΔT1 patterning was performed and is described in the methods and results, and a clear inner vs outer pattern was not clearly seen (examples in Figure 2). However, a new, comprehensive, lesion-by-lesion quantitative volumetric study could and should be done in the future, which will answer

your question.

Comment to the author

I appreciate the detailed response.

I accept that this analysis is beyond the scope of this paper, however, it would be helpful if you could add a brief comment on how susceptibility-weighted imaging (presence of paramagnetic RIM) could help elucidate whether changes in ΔT1 in NEL may be due to chronic disrupted BBB or due to chronic inflammation.

As you mention, enhancing lesions may show a transient paramagnetic RIM, however, here, you want to discriminate the possible underlying processes in the ΔT1 changes seen in NEL.

The authors have addressed my other comments.

Reviewer #2: The authors have addressed all the comments clearly and I recommend that the manuscript be accepted.

7. PLOS authors have the option to publish the peer review history of their article (what does this mean?). If published, this will include your full peer review and any attached files.

Reviewer #1: No

Reviewer #2: **Yes: **Aravinthan Varatharaj

---

## [Author Response · Author response to Decision Letter 1]

25 Mar 2021

Please see attached response to reviewer file

---

## [Editor Report · Decision Letter 2]

29 Mar 2021

Blood-brain barrier breakdown in non-enhancing multiple sclerosis lesions detected by 7-Tesla MP2RAGE ΔT1 mapping

PONE-D-20-33106R2

Dear Dr. Harrison,

We’re pleased to inform you that your manuscript has been judged scientifically suitable for publication and will be formally accepted for publication once it meets all outstanding technical requirements.

Kind regards,

Niels Bergsland

Academic Editor

PLOS ONE
---

## [Editor Report · Acceptance letter]

14 Apr 2021

PONE-D-20-33106R2 

Blood-brain barrier breakdown in non-enhancing multiple sclerosis lesions detected by 7-Tesla MP2RAGE ΔT_1_ mapping 

Dear Dr. Harrison:

I'm pleased to inform you that your manuscript has been deemed suitable for publication in PLOS ONE. Congratulations! Your manuscript is now with our production department. 

Kind regards, 

on behalf of

Dr. Niels Bergsland 

Academic Editor

PLOS ONE